# Shoring Up the Foundations: Fusing Model Embeddings and Weak Supervision

**Mayee F. Chen**[*1]    **Daniel Y. Fu**[*1]    **Dyah Adila**[2]    **Michael Zhang**[1]    **Frederic Sala**[2]    **Kayvon Fatahalian**[1]

**Christopher Ré**[1]

[1]Department of Computer Science , Stanford University, Stanford, CA, USA
[2]Department of Computer Science, University of Wisconsin-Madison, Madison, WI, USA

## Abstract

Foundation models offer an exciting new paradigm for constructing models with out-of-the-box embeddings and a few labeled examples. However, it is not clear how to best apply foundation models without labeled data. A potential approach is to fuse foundation models with weak supervision frameworks, which use weak label sources—pretrained models, heuristics, crowd-workers—to construct pseudolabels. The challenge is building a combination that best exploits the signal available in both foundation models and weak sources. We propose LIGER, a combination that uses foundation model embeddings to improve two crucial elements of existing weak supervision techniques. First, we produce finer estimates of weak source quality by partitioning the embedding space and learning per-part source accuracies. Second, we improve source coverage by extending source votes in embedding space. Despite the black-box nature of foundation models, we prove results characterizing how our approach improves performance and show that lift scales with the smoothness of label distributions in embedding space. On six benchmark NLP and video tasks, LIGER outperforms vanilla weak supervision by 14.1 points, weakly-supervised kNN and adapters by 11.8 points, and kNN and adapters supervised by traditional hand labels by 7.2 points.

## 1 INTRODUCTION

Foundation models—large pretrained models such as GPT-3, BERT, CLIP, and DALL-E [Brown et al., 2020, Devlin et al., 2019, Radford et al., 2021, Ramesh et al., 2021]—

offer powerful representations that can be used in a broad array of settings [Bommasani et al., 2021]. These models have achieved state-of-the-art performance on many tasks. However, it remains unclear how to best apply foundation models in situations where users lack access to any labeled data but do have some weak signals. These are the cases where another class of techniques—weak supervision [Ratner et al., 2018, Fu et al., 2020]—shines.

The broad success of foundation models (FMs) suggests that fusing them with weak supervision may offer substantial benefits. Intuitively, the signals present in both can be used to replace large amounts of hand-labeled data in supervised learning. These signals are complementary. Foundation models are trained on huge amounts of data and thus offer powerful general-purpose embeddings. Weak supervision frameworks rely on multiple weak sources of signal that can be synthesized into pseudolabels for downstream training. These weak sources typically express specialized domain expertise. The fusion may enable each component to be improved: FM embeddings can be used without labeled data, while weak sources may be extended to be more general-purpose.

Our goal is to combine these complementary signals to address two challenges in existing approaches to weak supervision. The first challenge is performing fine-grained estimation of source quality. Current weak supervision approaches typically coarsely model source quality by assuming error distributions are uniform over unlabeled points [Ratner et al., 2019, Fu et al., 2020], but source quality may vary across points in actuality. The second challenge is producing votes on points where sources abstain. Weak sources often abstain, so that current approaches suffer from low coverage and have many points lacking any signal. We seek to exploit the powerful embeddings from FMs—and the geometry induced by them—to address these challenges.

We propose LIGER, a new weak supervision approach based on the notion of *local* quality of weak sources in the FM embedding space (named after a well-known fusion of pow-

---

[*]Equal Contribution. A preliminary version of the results in this paper can be found at https://arxiv.org/abs/2006.15168.

*Accepted for the 38th Conference on Uncertainty in Artificial Intelligence (UAI 2022).*

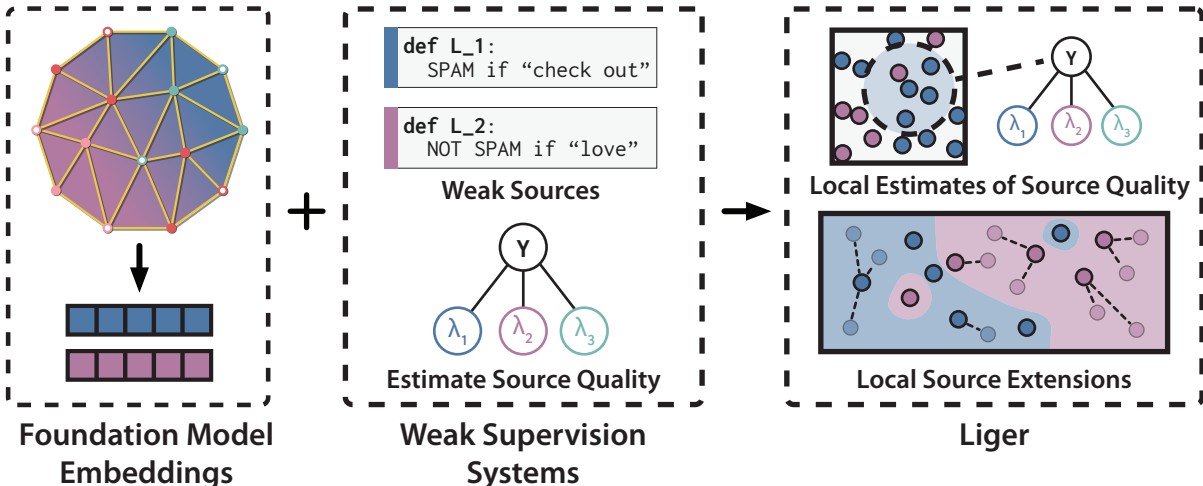



**Liger: Smart Fusion of Foundation Models and Weak Supervision**

```
def L_1:
    SPAM if "check out"
```

```
def L_2:
    NOT SPAM if "love"
```

**Weak Sources**

**Estimate Source Quality**

**Local Estimates of Source Quality**

**Local Source Extensions**

**Foundation Model Embeddings**

**Weak Supervision Systems**

**Liger**

Figure 1: LIGER fuses embeddings from foundation models (left) with weak supervision (middle) by exploiting local smoothness of the representations (right). LIGER uses the embeddings to a) produce more accurate local estimates of weak source quality (right, top), and b) to locally extend weak sources, improving their coverage (right, bottom).

erful animals). We introduce an efficient algorithm that partitions the embedding space and learns per-part local source accuracies. LIGER also extends weak sources into nearby regions of the embedding space that they previously abstained on, improving coverage. Despite the fact that FMs are typically black-box, our localized approach exploits a simple measurable notion of their signal: the smoothness of the label distribution in the embedding space. When the distribution of label values does not vary significantly over an embedding region, local source accuracies can be estimated well, and local source extensions maintain their accuracy. We introduce generalization error bounds that individually characterize the impact of partitioning and extending. These error bounds scale in the embedding smoothness and involve a bias-variance tradeoff in the number of partitions and the radii that specify extensions, suggesting that careful incorporation of the FM's signal into our approach is necessary.

We evaluate LIGER on six benchmark NLP and video supervision tasks, fusing weak sources with GPT-3 embeddings [Brown et al., 2020, Neelakantan et al., 2022] for the NLP tasks, and with image embeddings from CLIP [Radford et al., 2021] for the video tasks. We compare LIGER against using FMs or weak supervision on their own, as well as baseline techniques for fusing them together. First, LIGER outperforms two strong baselines for traditional supervision of FMs, kNN and adapters [Houlsby et al., 2019], by 7.2 points, and outperforms traditional weak supervision by 14.1 points. Next, LIGER outperforms kNN or adapter-based fusions of weak supervision and FMs by 11.8 points. We find that lift scales with embedding smoothness—confirming our theoretical findings. We measure the smoothness of CLIP embeddings against BiT-M [Kolesnikov et al., 2020], ResNet-101

embeddings pretrained on ImageNet [Russakovsky et al., 2015], and raw pixels on a video task. We find that CLIP embeddings are smoothest and result in the best performance. Similarly, we find that using the right prompt for GPT-3 has a strong effect on smoothness and performance on a relation extraction task.

In summary, we contribute:

- LIGER, a new approach for fusing foundation models with weak supervision by exploiting local smoothness of labels and weak sources in embedding space.

- Finite-sample generalization error bounds of our algorithm that scale in this smoothness.

- Evaluation of LIGER on six benchmark NLP and video weak supervision tasks, where LIGER outperforms simple fusions of foundation models and weak supervison, as well as either on its own.

## 2 BACKGROUND

We describe the problem setting for weak supervision (Section 2.1). We introduce two general challenges in weak supervision that our approach using foundation model embeddings can mitigate. We then propose a model and explain its two stages—source quality estimation and pseudolabel inference (Section 2.2). We provide a brief background on the estimation technique from Fu et al. [2020], on top of which we build our approach.

## 2.1 PROBLEM SETUP

Our goal is to predict label $y \in \mathcal{Y} = \{-1, +1\}$ from datapoints $x \in \mathcal{X}$. If we had access to pairs $(x, y)$, we could train a supervised model. However, we do not have access to any samples of $y$; instead, we observe $m$ *weak sources* $\boldsymbol{\lambda} = \{\lambda_1, \ldots, \lambda_m\}$, each voting or abstaining on each point $x$ via a probabilistic *labeling function* $\lambda_j : \mathcal{X} \to \mathcal{Y} \cup \{0\}$ for all $j \in [m]$. We refer to $\lambda_j(x) = 0$ as an abstain, which occurs when a source is uncertain or not applicable on a point.

We also have access to FM embeddings. These embeddings are the outputs of a mapping $f : \mathcal{X} \to \mathcal{Z}$ from input space to an embedding space $\mathcal{Z}$ equipped with metric $\rho : \mathcal{Z} \times \mathcal{Z} \to \mathbb{R}^+$. This mapping is fixed and obtained from an off-the-shelf model. Overall, we have an unlabeled dataset $\mathcal{D} = \{x_i\}_{i=1}^n$ of $n$ i.i.d. points, as well as access to $m$ weak sources and the embedding map $f$.

Given an input $x$ and $\boldsymbol{\lambda}(x)$, we aim to learn a *label model* that predicts $y$ by estimating $\hat{\Pr}(y|\boldsymbol{\lambda}, x)$ (we drop the $x$ in $\boldsymbol{\lambda}(x)$ when obvious). The goal of the label model is to combine sources based on their individual accuracies (i.e. $\lambda_i$'s rate of agreement with $y$) by upweighting high-quality sources and downweighting low-quality ones. The resulting pseudolabels given by $\hat{\Pr}(y|\boldsymbol{\lambda}, x)$ can be used to train a downstream supervised *end model* or used just directly as predictions. The latter case is often ideal, since users need not train an additional model. We focus on this setting.

**Two Challenges and Opportunities.** Next, we describe two challenges common to weak supervision techniques. Fusing weak supervision with FM embeddings presents opportunities to mitigate these challenges.

- **Coarse Accuracy Modeling.** The most common assumption in weak supervision is to model $\hat{\Pr}(y|\boldsymbol{\lambda}, x)$ as $\hat{\Pr}(y|\boldsymbol{\lambda})$. That is, conditioned on the weak sources, the true label is viewed as independent of the features, so only one set of accuracies is learned over the data. Removing this assumption is desirable, since the feature space may have information about the task not captured fully by weak sources. However, naively attempting to model per-point accuracies leads to noisy estimation.

- **Low Coverage.** Weak sources frequently abstain, leading to low coverage—a situation where much of the dataset has no votes. A simple mitigation is to extend votes from nearby non-abstaining points, but this is risky if the notion of distance is not well-aligned with the label distribution.

An intuitive way to tackle these two challenges is to operate *locally*. Suppose the source votes and the true label satisfy some level of smoothness such that within some local region of the feature space, they have a low probability of changing values. We can then model accuracies specific to such local regions and can extend source votes to points they abstain on within the regions. However, raw image and text features

may lack signal and not offer sufficient smoothness to permit operating locally. By acting on the embedding space, the desired smoothness property is improved (see Figure 2). We can thus obtain finer-grained accuracy estimation and improved coverage by using FM embeddings to model local accuracies and extend locally.

Next, we make these notions concrete by presenting the explicit model for $\Pr(y, \boldsymbol{\lambda}|x)$.

## 2.2 LABEL MODEL

We model $\Pr(y, \boldsymbol{\lambda}|x)$ as a probabilistic graphical model. Our use of this model has two steps. First, in training, we must estimate the accuracy parameters of $\Pr(y, \boldsymbol{\lambda}|x)$ without access to $y$. Then, at inference, we compute $\hat{\Pr}(y|\boldsymbol{\lambda}, x)$.

Let the graphical model be based on $G = (V, E)$, where $V = y \cup \boldsymbol{\lambda}$ and $E$ consists of edges from $y$ to each $\lambda_j$ (see Figure 1 middle). For simplicity, we assume there are no dependencies between the weak sources, although the dependencies can be learned [Varma et al., 2019] and handled by our choice of base estimator from [Fu et al., 2020]. Therefore, our approach can be extended to that case as well. We model the data distribution as

$$\Pr(y, \boldsymbol{\lambda}|x) = \frac{1}{Z} \exp\Big( \underbrace{\theta_y(x)y}_{\text{Class Balance}} + \sum_{i=1}^m \underbrace{\theta_i(x)\lambda_i y}_{\text{Source Accuracy}}$$
$$+ \sum_{i=1}^m \underbrace{\theta_{i,0}(x)\mathbb{1}\{\lambda_i = 0\}}_{\text{Abstain Rate}} \Big) \quad (1)$$

with partition function $Z$ and a set of canonical parameters per $x$, $\Theta(x) = \{\theta_y(x), \theta_i(x), \theta_{i,0}(x) \; \forall i \in [m]\}$. An important property above is that $\lambda_i \perp\!\!\!\perp \lambda_j | y, x \; \forall i, j \in [m]$.

The model concretely portrays the two challenges in weak supervision. First, canonical parameters $\Theta(x)$ that are a function of the input can capture varying accuracy across the data. This is less strict than prior formulations that model the marginal $\Pr(y, \boldsymbol{\lambda})$ with one set of canonical parameters without considering input data. However, estimating $\Theta(x)$ is challenging; parametric approaches require certain assumptions on the function $\Theta$ as well as the distribution of $x$ in order to recover the ground truth labels, but these assumptions (e.g., Gaussian $x$) are often not realistic. Standard nonparametric approaches have a high computational complexity and rely on smoothness of the input space $\mathcal{X}$. Second, when $\lambda_i(x) = 0$, the weak source provides no information on $x$ at inference and is thus typically ignored on that point in previous approaches. This is reflected in the graphical model by Lemma 2 in Appendix C.1, by which $\Pr(y|\lambda_i = 0, \boldsymbol{\lambda} \backslash \lambda_i, x) = \Pr(y|\boldsymbol{\lambda} \backslash \lambda_i, x)$. In fact, the weak sources provide no direct signal on $x$ when $\boldsymbol{\lambda}(x) = \vec{0}$.

**Pseudolabel Inference.** To perform inference, we compute $\hat{\Pr}(y|\boldsymbol{\lambda}, x)$ for some $x \in \mathcal{X}$. This is done via Bayes'

rule and the conditional independence of weak sources: $\Pr(y|\boldsymbol{\lambda}, x) = \prod_{i=1}^{m} \Pr(\lambda_i|y, x) \Pr(y|x) / \Pr(\boldsymbol{\lambda}|x)$. The latent parameter of interest in this decomposition is $\Pr(\lambda_i|y, x)$, which corresponds to the accuracy of $\lambda_i$.

**Source Parameter Estimation.** Previous approaches have considered how to estimate $\Pr(\lambda_i|y)$ in a model of $\Pr(\boldsymbol{\lambda}, y)$ via the *triplet method* [Fu et al., 2020], using conditional independence properties. For our setting, (1) tells us that $\lambda_i y \perp\!\!\!\perp \lambda_j y | \lambda_i \wedge \lambda_j \neq 0, x$ for any $i \neq j$ (Lemma 3 in Appendix C.1). As a result, $\mathbb{E}\left[\lambda_i y | \lambda_i \neq 0, x\right] \times \mathbb{E}\left[\lambda_j y | \lambda_j \neq 0, x\right] = \mathbb{E}\left[\lambda_i \lambda_j y^2 | \lambda_i \wedge \lambda_j \neq 0, x\right] = \mathbb{E}\left[\lambda_i \lambda_j | \lambda_i \wedge \lambda_j \neq 0, x\right]$, which consists of observable variables. Define $a_i(x) = \mathbb{E}\left[\lambda_i y | \lambda_i \neq 0, x\right]$ as the *accuracy* of $\lambda_i$ on $x$. If we introduce a third $\lambda_k$, we can generate a system of equations over $a_i(x), a_j(x), a_k(x)$ in terms of the conditional expected products of pairs of $\lambda_i, \lambda_j, \lambda_k$. As a result,

$$|a_i(x)| := \qquad\qquad\qquad\qquad (2)$$
$$\sqrt{\left|\frac{\mathbb{E}\left[\lambda_i \lambda_j | \lambda_i \wedge \lambda_j \neq 0, x\right] \mathbb{E}\left[\lambda_i \lambda_k | \lambda_i \wedge \lambda_k \neq 0, x\right]}{\mathbb{E}\left[\lambda_j \lambda_k | \lambda_j \wedge \lambda_k \neq 0, x\right]}\right|},$$

and likewise for $\hat{a}_j(x), \hat{a}_k(x)$. More details are in Appendix C.2. (1) allows us to write $\Pr(\lambda_i|y, x) = \frac{1+\mathrm{sgn}(\lambda_i y)a_i(x)}{2} \times \Pr(\lambda_i \neq 0|x)$ (Lemmas 2 and 4), so the desired probability estimate is just a linear transformation of $a_i(x)$ scaled by $\lambda_i$'s coverage.

# 3 FUSION ALGORITHM

We are ready to present LIGER, our approach to fusing foundation model embeddings and weak supervision. We explain the two components: first, how to compute conditional estimates of the label model parameters over local regions of the partitioned embedding space for finer-grained accuracy estimation; second, how to extend weak sources via a kNN-like augmentation in the embedding space, improving their coverage and hence the signal available at inference. The full approach is shown in Algorithm 1.

**Local Parameter Estimation** Our first task is to compute the label model's local parameters. Based on (2), the quantities to estimate are of the form $\mathbb{E}\left[\lambda_i \lambda_j | \lambda_i \wedge \lambda_j \neq 0, x\right]$, $\Pr(\lambda_i \neq 0|x)$, $\Pr(\boldsymbol{\lambda}|x)$, $\Pr(y|x)$. These conditional statistics can be estimated using nonparametric approaches such as the Nadaraya-Watson estimator, but they require $\mathcal{O}(n)$ computations per point at inference.

Instead of estimating parameters per point, we partition the embedding space and compute *per-part* statistics. Intuitively, this choice exploits smoothness. If label distributions are smooth, i.e., they do not vary greatly within a local region, it is sufficient to estimate per-point statistics using a part given that parts are not too large. Controlling the size of the

---

**Algorithm 1** LIGER

**Input:** Dataset $\mathcal{D} = \{x_i\}_{i=1}^{n}$, weak sources $\boldsymbol{\lambda}$, embedding mapping $f$ and metric $\rho$, threshold radii $r_1, \ldots r_m$, partition $\mathcal{C}$ and class balances $\Pr(y|C_j)$ for $j \in [s]$.
**Returns:** Label model $\hat{\Pr}(y|\bar{\boldsymbol{\lambda}}, x)$.
**for** $\lambda_i \in \boldsymbol{\lambda}$ **do**
    Construct extended source $\bar{\lambda}_i$ using $r_i, f, \rho$ as in (3).
**end for**
**for** $C_j \in \mathcal{C}$ **do**
    **for** $\bar{\lambda}_i \in \bar{\boldsymbol{\lambda}}$ **do**
        Compute accuracy $\hat{a}_i(C_j)$ using Algorithm 2 on $\bar{\lambda}_i$ over $C_j$, and compute coverage $\hat{\Pr}(\bar{\lambda}_i \neq 0|C_j)$ on $\mathcal{D}$.
        Set $\hat{\Pr}(\bar{\lambda}_i|y, C_j)$ equal to $\frac{1+\mathrm{sgn}(\bar{\lambda}_i y)\hat{a}_i(C_j)}{2}\hat{\Pr}(\bar{\lambda}_i \neq 0|C_j)$ for $\bar{\lambda}_i \in \{-1, 1\}$, $\hat{\Pr}(\bar{\lambda}_i = 0|C_j)$ otherwise.
    **end for**
    Compute $\hat{\Pr}(\bar{\boldsymbol{\lambda}}|C_j)$ on $\mathcal{D}$.
**end for**
**return** For test point $x \in \mathcal{X}$, compute $\hat{\Pr}(y|\bar{\boldsymbol{\lambda}}, x) = \hat{\Pr}(y|\bar{\boldsymbol{\lambda}}, C(x)) = \frac{\prod_{i=1}^{m} \hat{\Pr}(\bar{\lambda}_i|y, C(x)) \Pr(y|C(x))}{\hat{\Pr}(\bar{\boldsymbol{\lambda}}|C(x))}$.

---

partition is thus important in determining how well we can approximate per-point statistics.

Concretely, partition $\mathcal{Z}$ into $s$ subsets $\mathcal{C} = \{C_1, \ldots, C_s\}$ of equal size $n' = \frac{n}{s}$ (we use K-means clustering with $K = s$ in practice). Denote $C(x)$ as the subset $f(x)$ belongs to. Instead of estimating statistics and performing inference conditioned on $x$, we condition on $C(x)$, producing $s$ sets of parameters overall. We estimate $\mathbb{E}\left[\lambda_i \lambda_j | \lambda_i \wedge \lambda_j \neq 0, C(x)\right]$, yielding a local accuracy estimate $\hat{a}_i(C(x))$ formalized in Algorithm 2, as well as $\Pr(\lambda_i \neq 0|C(x))$, $\Pr(\boldsymbol{\lambda}|C(x))$, $\Pr(y|C(x))$. Then, we use $\hat{\Pr}(y|\boldsymbol{\lambda}, x) = \hat{\Pr}(y|\boldsymbol{\lambda}, C(x))$ as our label model prediction on $x$. These estimates are done over the subsets; for instance, $\Pr(\boldsymbol{\lambda}|C(x)) \approx \frac{1}{n'}\sum_{x' \in C(x)} \mathbb{1}\{\boldsymbol{\lambda}(x')=\boldsymbol{\lambda}\}$. We assume that class balance on subsets, $\Pr(y|C(x))$, are known. There are also several techniques that can be used to estimate these [Ratner et al., 2019], or they can be treated as hyperparameters.

**Weak Source Extension** Next, we improve the model of $\hat{\Pr}(y|\boldsymbol{\lambda}, x)$ by increasing source coverage. Let $\bar{\lambda}_i$ be an extended labeling function with corresponding threshold radius $r_i > 0$ for $i \in [m]$. The extension works as follows. For any $x$, let $\mathrm{NN}(x) = \mathrm{argmin}_{x' \in \mathcal{D}:\lambda_i(x) \neq 0} \rho(f(x), f(x'))$ be the nearest neighbor of $x$ in embedding space from $\mathcal{D}$ such that $\lambda_i$ has coverage on it. $\bar{\lambda}_i$ uses nearest neighbors to weakly label points within $r_i$ of $\lambda_i$'s support on $\mathcal{D}$. Formally,

$$\bar{\lambda}_i(x) := \begin{cases} \lambda_i(x) & \lambda_i(x) \neq 0 \\ \lambda_i(\mathrm{NN}(x)) & \rho(f(x), f(\mathrm{NN}(x))) \leq r_i \\ 0 & \text{o.w.} \end{cases} \quad (3)$$

We can view $\bar{\lambda}_i$ as an augmentation on $\lambda_i$ using $\mathcal{D}$ and $f$. We thus perform parameter estimation and inference using $\bar{\boldsymbol{\lambda}}$ instead of $\boldsymbol{\lambda}$, namely learning $\Pr(y|\bar{\boldsymbol{\lambda}}, C(x))$.

There are two advantages to using extended sources. First, extended sources improve sampling error, since expressions like $\mathbb{E}\left[\lambda_i \lambda_j | \lambda_i \wedge \lambda_j \neq 0, x\right]$ are estimated over more data in $\mathcal{D}$. Second, $\bar{\lambda}_i$ provides signal at inference on points that $\lambda_i$ previously abstains on. However, the quality of this signal greatly depends on $r_i$. If $\lambda_i$ is overextended and the embedding space is not sufficiently smooth, points far away from $\lambda_i$'s support may receive incorrect extended source votes, suggesting that careful choice of $r_i$ is needed.

Our approach combines the two components discussed—partitioning the embedding space and extending sources—to output predictions $\hat{\Pr}(y|\bar{\boldsymbol{\lambda}}, C(x))$ as in Algorithm 1. Note that our approach builds on the algorithm from Fu et al. [2020], but partitioning and extending can also be done on top of other weak supervision algorithms that model things differently.

# 4 THEORETICAL ANALYSIS

Now we turn to analyzing Algorithm 1. Our goal is to understand how performance depends on the key parameters: fineness of the partition $\mathcal{C}$, radii $r_i$ of the extensions used to improve coverage, and smoothness of the embedding space.

We begin with a result on the generalization error of the label model $\hat{\Pr}(y|\boldsymbol{\lambda}, x)$, which relies on the number of partitions $s$ to control the granularity of the estimated parameters (Theorem 1). Then, we discuss the improvement from using $\bar{\boldsymbol{\lambda}}$ instead of $\boldsymbol{\lambda}$. We first bound the local accuracy of an extended source in a region it previously abstains (Lemma 1), and then we show that as long as this local accuracy is better than random, we can further reduce the generalization error (Theorem 2). The former result presents a bias-variance tradeoff depending on $s$, while the latter has a tradeoff dependent on the threshold radius $r_i$. In both cases, $s$ and $r_i$ must be carefully set based on the signal in the FM embeddings, namely the smoothness of label distributions in the FM embedding space, in order to optimize performance. We provide proofs in Appendix D, synthetic experiments supporting our findings in Appendix F.3, and smoothness measurements on real data in Section 5.2 and Appendix F.2.

Define the generalization error of the label model using weak sources $\boldsymbol{\lambda}$ as the expected cross-entropy loss, $L(\boldsymbol{\lambda}) = \mathbb{E}_{\mathcal{D},x,y,\boldsymbol{\lambda}}[-\log \hat{\Pr}(y|\boldsymbol{\lambda}, x)]$.

## 4.1 LABEL MODEL GENERALIZATION ERROR

We bound the generalization error $L(\boldsymbol{\lambda})$ of the label model using the unextended, initial weak sources. The key quantity in this analysis is embedding smoothness:

**Definition 1** (Lipschitzness). *The distributions* $\Pr(y|x)$ *and* $\Pr(\lambda_i|y, x)$ *are Lipschitz-smooth on the metric space* $(\mathcal{Z}, \rho)$ *with constants* $K_y, K_\lambda, K_{\lambda,0} > 0$ *if for all* $i \in [m]$,

$$|\Pr(y = 1|x) - \Pr(y = 1|x')| \leq K_y \rho(f(x), f(x')),$$
$$|\Pr(\lambda_i = 1|y, \lambda_i \neq 0, x) - \Pr(\lambda_i = 1|y, \lambda_i \neq 0, x')|$$
$$\leq K_\lambda \rho(f(x), f(x')),$$
$$|\Pr(\lambda_i \neq 0|x) - \Pr(\lambda_i \neq 0|x')| \leq K_{\lambda,0} \rho(f(x), f(x')),$$

*We refer to these three properties as label, source, and coverage Lipschitzness, respectively.*

In words, if the constants $K_y, K_\lambda, K_{\lambda,0}$ are small, the class balance of $y$ and the way each source votes (or doesn't) do not vary significantly over a local region of the embedding space.

We define some additional quantities. Set $\alpha = \max_i \mathbb{E}_x \left[\frac{1}{p_{ij}} \mid p_{ij} \neq 0\right]$, where $p_{ij} = \Pr(\lambda_i \neq 0|f(x) \in C_j)$ is the coverage of $\lambda_i$ on $C_j$, to be the largest average inverse source coverage over the subsets. $\alpha$ corresponds to how often sources abstain. Assume that $a_i(C_j) > 0$ for all $\lambda_i$ and $C_j$, meaning that the average source accuracy on a subset is better than random. Then, define $a_{\max} = \max_{i,j} a_i(C_j)$, and $b_{\min} = \min_{i,j,k}\{\mathbb{E}\left[\lambda_i \lambda_k | \lambda_i \wedge \lambda_k \neq 0, C_j\right], \hat{\mathbb{E}}\left[\lambda_i \lambda_k | \lambda_i \wedge \lambda_k \neq 0, C_j\right]\}$ as the minimum rate of agreement between sources over subsets, where $\hat{\mathbb{E}}$ denotes the empirical estimate on $\mathcal{D}$. Define $d_{C_j} = \max_{f(x),f(x') \in C_j} \rho(f(x), f(x'))$ as the diameter of $C_j$ and $d_{\mathcal{C}} = \mathbb{E}_x\left[d_{C(x)}\right]$ as its average.

**Theorem 1.** *Suppose that data* $x, y, \boldsymbol{\lambda}$ *follows the model in* (1) *and* $\Pr(y|x)$ *and* $\Pr(\lambda_i|y, x)$ *for each* $\lambda_i$ *are Lipschitz-smooth. The generalization error of the label model* $\hat{\Pr}(y|\boldsymbol{\lambda}, x)$ *in Algorithm 1 when* $r_i = 0 \ \forall i$ *can be decomposed into* $L(\boldsymbol{\lambda})$=*Bias*+*Variance*+*Irreducible Error*+ $o(1/n)$*, where*

$$Bias \leq 2d_{\mathcal{C}}(K_y + mK_\lambda + mK_{\lambda,0}),$$
$$Variance \leq \frac{ms}{n}\left(\frac{3\alpha(1 - b_{\min}^2)}{8b_{\min}^2(1 - a_{\max}^2)}\left(\frac{1}{b_{\min}^4} + \frac{2}{b_{\min}^2}\right) + 1\right),$$
$$Irreducible\ Error = H(y|\boldsymbol{\lambda}, x),$$

*where* $H(y|\boldsymbol{\lambda}, x)$ *denotes conditional entropy.*

We discuss each term of this bound.

- The bias comes from the partition $\mathcal{C}$, since conditional statistics on $C(x)$ are not equivalent to those on $x$. When the embedding space is smooth with small $K_y, K_\lambda, K_{\lambda,0}$, the bias is low. Note that making the subset diameter $d_C \to 0$ makes the bias go to zero.
- The variance comes from sampling error in Algorithm 2 and $\hat{\Pr}(\lambda_i \neq 0|C_j)$. This quantity scales in $\mathcal{O}(s\alpha/n)$ and also depends on accuracy and agreement among weak sources.

- The irreducible error depends on quality of $\boldsymbol{\lambda}$. If knowledge of $\boldsymbol{\lambda}$ significantly reduces uncertainty in $y$, i.e., the sources contain lots of signal, this quantity is low. On the other hand, $H(y|\boldsymbol{\lambda}, x)$ is maximized when $\boldsymbol{\lambda} \perp\!\!\!\perp y|x$, i.e. there is no signal about $y$ in $\boldsymbol{\lambda}$.

Our result reveals a bias-variance tradeoff dependent on the number of parts $s$. As $s$ increases, subset diameter $d_{\mathcal{C}}$ tends to decrease, resulting in lower bias because the subset parameters estimated will be closer in true value to those conditional on $x$. The variance increases in $s$ because there are fewer points per subset for estimation. The $s = 1$ case, which incurs a large bias, is algorithmically equivalent to the approach in Fu et al. [2020]. Such approaches thus suffer from model misspecification in our setting—and likely in most practical cases—as they assume uniform quality per source.

## 4.2 IMPROVEMENT FROM EXTENSIONS

Suppose that $x, y, \bar{\boldsymbol{\lambda}}$ follows (1). When we use $\bar{\boldsymbol{\lambda}}$ rather than $\boldsymbol{\lambda}$ (i.e. $r_i \neq 0$), there are several changes to the decomposition in Theorem 1:

- The bias is now bounded by $2d_{\mathcal{C}}K_y + 2m(d_{\mathcal{C}} + 2\max_i r_i)(K_\lambda + K_{\lambda,0})$ (see Lemma 8 in Appendix D). We must consider when $NN(x)$ is not in $C(x)$, essentially resulting in a wider subset diameter.
- The variance is still $\mathcal{O}(1/n)$, but multiplicative factors change. For instance, $\alpha$ decreases due to improved coverage, thus decreasing the variance.
- The irreducible error is now $H(y|\bar{\boldsymbol{\lambda}}, x)$.

We analyze $H(y|\bar{\boldsymbol{\lambda}}, x)$ in this section. $\bar{\lambda}_i$ provides more signal than $\lambda_i$ at inference on points where $\lambda_i(x) = 0$, but the signal about $y$'s value may be incorrect. Extending $\lambda_i$ using too large of $r_i$ could yield incorrect source votes, resulting in lower accuracy of the extended weak source.

We first present a result on how $r_i$ controls the extended source's accuracy. Define $a_i = \mathbb{E}\left[\lambda_i y | \lambda_i \neq 0\right]$ as the average accuracy of $\lambda_i$, and $\bar{a}_i(r_i) = \mathbb{E}\left[\bar{\lambda}_i y | \bar{\lambda}_i \neq 0, \lambda_i = 0\right]$ as $\bar{\lambda}_i$'s average accuracy on the extended region. We also need a notion of smoothness of $y$ between the original support and the extended region. We define a local notion of *probabilistic Lipschitzness* (PL), originally introduced in Urner and Ben-David [2013].

**Definition 2** (Probabilistic Lipschitzness). *Define* $\mathcal{P}_{\lambda_i} = \Pr_{x,y}(\cdot | \lambda_i \neq 0)$ *to be the distribution of* $(x, y)$ *over the support of* $\lambda_i$, *and let* $\mathcal{P}_{\lambda_i, x}$ *be its marginal distribution on $x$. Then* $\mathcal{P}_{\lambda_i}$ *is* $M$-*probabilistically Lipschitz for an increasing function* $M : \mathbb{R}^+ \to [0, 1]$ *if for any* $r > 0$,

$$\Pr_{x,y \sim \mathcal{P}_{\lambda_i}} (\exists(x', y') \in \mathcal{X} \backslash supp(\mathcal{P}_{\lambda_i, x}) \times \mathcal{Y} :$$
$$\rho(f(x), f(x')) \leq r, y' \neq y) \leq M(r).$$

*We refer to this property as local label PL.*

In words, the probability that there is a point outside of the support of $\lambda_i$ but within $r$ of $(x, y) \sim \mathcal{P}_{\lambda_i}$ with a different label from $y$ is bounded by an increasing function of $r$. We also define $\beta_i = \mathbb{E}[\lambda_i y | \lambda_i \neq 0, \exists(x', y') : \lambda_i(x') = 0, \rho(f(x), f(x')) \leq r_i, y' = y]$ as $\lambda_i$'s accuracy over a region close to where $\lambda_i$ is extended and $y$ changes value.

With this definition, we show that:

**Lemma 1.** *Suppose* $\mathcal{P}_{\lambda_i}$ *is* $M$-*probabilistically Lipschitz. The average accuracy of* $\bar{\lambda}_i$ *on the extended region is at least* $\bar{a}_i(r_i) \geq a_i - (1 + \beta_i)M(r_i)$.

Our result provides local accuracy guarantees on $\bar{\lambda}_i$ as a function of the original $\lambda_i$'s accuracy, the probabilistic Lipschitzness of the embedding space, and the $r_i$ the user sets. Extending a source with higher original accuracy will yield stronger accuracy guarantees in the extended region. On the other hand, if $M(r_i)$ is too large due to improper $r_i$ or lack of smoothness, the true label is more likely to change value, and hence accuracy in the extended region worsens.

Now we can use our result on $\bar{a}_i(r_i)$ to analyze the improvement in irreducible error. We extend just one weak source $\lambda_i$ by $r_i$ and keep $\lambda_{-i} := \boldsymbol{\lambda} \backslash \lambda_i$ unextended. Define $p_i = \Pr(\bar{\lambda}_i \neq 0, \lambda_i = 0)$ as the proportion of the region where $\bar{\lambda}_i$ is extended and $p(\lambda_{-i}) = \mathbb{E}_{y', \lambda_{-i}, \bar{\lambda}_i \neq 0, \lambda_i = 0}\left[\Pr(y = y' | \lambda_{-i}, x)\right]$ as the label model's probability of outputting the correct label in the extension region when only using $\lambda_{-i}$.

**Theorem 2.** *Suppose that data follows the model in* (1). *The irreducible error decreases by at least the following amount when using* $\bar{\lambda}_i$ *rather than* $\lambda_i$ *in Algorithm 1:*

$$H(y|\boldsymbol{\lambda}, x) - H(y|\bar{\boldsymbol{\lambda}}, x) \geq 2p_i(1 - p(\lambda_{-i}))^2 \cdot \bar{a}_i(r_i)^2.$$

Lift increases with probability mass $p_i$ on the extended region since more of the data is impacted by $\bar{\lambda}_i$. Lift is not as significant if $p(\lambda_{-i})$ is large because the other weak sources already are providing sufficient signal for $y$. Most importantly, lift scales with how far $\bar{a}_i(r_i)$ is from 0 (random voting). This highlights a tradeoff in $r_i$: as $r_i$ increases, $p_i$ increases but the lower bound on $\bar{a}_i(r_i)$ from Lemma 1 decreases. This shows that threshold radii must be selected carefully; if the embedding space has strong probabilistic Lipschitzness (i.e. small $M$) or the original weak source has high accuracy, then the source can be extended further while providing lift. However, overextension of the source can yield low local accuracy and thus less lift.

Our results demonstrate that $s$ and $r_i$ control the label model's performance, and setting these terms depends on how smooth label distributions are in the embedding space.

# 5 EXPERIMENTS

This section evaluates the following claims about LIGER:

- **Performance (Section 5.1):** LIGER outperforms vanilla weak supervision, as well as baseline approaches for using foundation models directly, either with traditional weak supervision or hand supervision.

- **Smoothness (Section 5.2):** Lift is correlated with the smoothness of the label distribution in the representation space. We measure smoothness and performance of CLIP against three other embedding methods on a video task, and measure three prompting strategies for GPT-3 on a relation extraction task.

- **Ablations (Section 5.3):** Both components of LIGER—partitioning the representation space and extending labeling function votes—are important for performance.

**Datasets** We evaluate LIGER on six benchmark NLP and video tasks used to evaluate previous weak supervision methods [Fu et al., 2020, Zhang et al., 2021]. In NLP, **Spam** identifies spam YouTube comments [Alberto et al., 2015]; **Weather** identifies the sentiment of weather-related tweets [Cro]; and **Spouse** identifies spouse relationships in newspaper articles [Corney et al., 2016]. In video, **Commercial** identifies commercial segments in TV news [Hong et al., 2021, Fu et al., 2019]; **Tennis** identifies rallies in tennis segments; and **Basketball** identifies basketball videos in a subset of ActivityNet [Caba Heilbron et al., 2015]. Each dataset consists of a large unlabeled training set, a smaller hand-labeled *development set* (train/dev split sizes from 187/50 points to 64,130/9,479 points), and a held-out test set. We use the unlabeled training set to train label models and use the development set for a) training of traditional supervision baselines, and b) hyperparameter tuning of the label models, including $s$ and $r_i$.

**Pre-trained embeddings** For the NLP datasets, we use pre-trained GPT-3 [Brown et al., 2020] embeddings from OpenAI's Ada model. For **Spam** and **Weather**, we simply embed the text directly. For **Spouse**, we add a prompt "Are [person 1] and [person 2] spouses?" after the end of the sentence. We discuss further prompting strategies in Section 5.2. For video datasets, we use image embeddings from CLIP [Radford et al., 2021] over individual frames of the videos.

## 5.1 PERFORMANCE

We compare LIGER against baseline approaches for fusing foundation models with weak supervision, as well as against using either on their own. We split our evaluation into two parts: methods that only have access to weak sources, and methods that additionally have access to the dev set.

**Weak Sources Only** We compare the performance of LIGER against vanilla weak supervision's label model (WS-LM) [Fu et al., 2020], as well as two end models, weakly-supervised kNN (WS-kNN), and weakly-supervised adapters (WS-Adapter). In the latter two methods, we use the predictions from WS-LM to generate pseudolabels for the train set and use the FM embeddings as input data (since we do not access the full FM) to the kNN and adapter approaches. We consider an adapter that is a linear layer on the FM embeddings. We also provide results on 3-layer MLP adapters in Appendix F.

Table 1 (left) shows the results, as well as statistics on the additive change in coverage (% of the dataset that sources vote on) between LIGER and WS-LM. LIGER outperforms WS-LM and has better coverage (33.2 points on average). LIGER also outperforms both of the baseline approaches for fusing foundation models with weak supervision, WS-kNN and WS-Adapter.

**Weak Sources and Dev Labels** Next, we compare performance against methods that have access to a small hand-labeled dev set. We compare against two baselines: kNN and Adapter, both trained over the dev set labels. For our method LIGER-Adapter, we train an adapter over LIGER labels on the train set, as well as the dev labels. In some cases, LIGER labels are too noisy to provide good signal on the train set; in this case, our solution automatically downsamples the pseudolabels on the train set. We also provide the original LIGER prediction as input to the adapter. See Appendix E for the details.

Table 1 (right) shows the results. LIGER-Adapter outperforms Adapter and kNN. On the datasets where LIGER labels are very accurate, we see additional lift from the adapters because we have more points to train on. When the labels are not very accurate, our downsampling prevents the noisy labels from harming adapter performance. In one case, learning an adapter over the embeddings is very hard (**Spouse**). Here, providing the LIGER prediction as input is critical for performance.

## 5.2 EMBEDDING SMOOTHNESS

We measure how smoothness of the embedding space affects the performance of LIGER. First, we compare embeddings from CLIP against BiT-M embeddings [Kolesnikov et al., 2020], a ResNet-101 pretrained on ImageNet [Russakovsky et al., 2015], and raw pixels. Second, we vary the GPT-3 prompting strategy for **Spouse** and compare against two alternative methods that result in a less smooth representation. We report label Lipschitzness—the smoothness of embeddings with respect to ground-truth labels—in this section. See Appendix F.2 for additional measures of Lipschitzness.

Figure 2 (left) shows the performance of CLIP, BiT-M,

| | Task | Weak Sources Only | | | | | Dev Labels Available | | |
|---|---|---|---|---|---|---|---|---|---|
| | | WS-kNN | WS-Adapter | WS-LM | LIGER | △Coverage | kNN | Adapter | LIGER-Adapter |
| NLP | Spam | 72.8 | 92.3 | 83.6 | **95.0** | +45.5 | 91.2 | 94.4 | **95.4** |
| | Weather | 62.0 | 86.0 | 78.0 | **98.0** | +90.2 | 92.0 | 90.0 | **96.8** |
| | Spouse | 16.9 | 17.1 | 47.0 | **52.2** | +12.1 | 21.6 | 15.7 | **49.6** |
| Video | Basketball | 33.3 | 48.9 | 27.9 | **69.6** | +8.3 | 64.4 | 79.3 | **79.5** |
| | Commercial | 84.7 | 92.8 | 88.4 | **93.5** | +18.8 | 92.0 | 93.0 | **93.2** |
| | Tennis | 83.0 | **83.8** | 82.0 | 83.3 | +32.5 | 73.2 | 83.1 | **84.0** |

Table 1: Left: LIGER performance compared to baselines that only have access to weak labels, as well as the change in coverage from traditional weak supervision. Right: LIGER-Adapter performance compared to baselines that have access to dev labels. Scores are F1 except for Spam and Weather (accuracy); best score in bold in each setting.

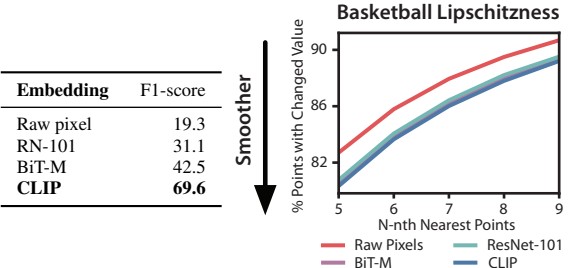

| Embedding | F1-score |
|---|---|
| Raw pixel | 19.3 |
| RN-101 | 31.1 |
| BiT-M | 42.5 |
| **CLIP** | **69.6** |

| Prompting | F1-score |
|---|---|
| No Prompt | 48.5 |
| Prompt Beginning | 50.2 |
| **Prompt End** | **52.2** |

Figure 2: Left: LIGER performance and smoothness measurements of CLIP, BiT-M, ResNet-101, and raw pixels as embeddings for **Basketball**. Right: LIGER performance and smoothness measurements of no prompting, prompting at beginning, and prompting at end in GPT-3 for **Spouse**.

ResNet-101, and raw pixels as embeddings for LIGER, as well as measures of Lipschitzness for each method (lower is smoother). CLIP embeddings are smoother than the other methods—which matches their performance when used in LIGER.

| Task | LIGER ($s$) | -Part | -Ext ($s$) | -Part, -Ext |
|---|---|---|---|---|
| **Spam** | **95.0** (2) | 94.0 | 92.0 (7) | 83.6 |
| **Weather** | **98.0** (3) | 96.0 | 94.0 (5) | 78.0 |
| **Spouse** | **52.2** (6) | 50.0 | 49.3 (5) | 47.0 |
| **Basketball** | **69.6** (2) | **69.6** | 21.9 (2) | 27.9 |
| **Commercial** | **93.5** (3) | 92.3 | 91.4 (5) | 88.4 |
| **Tennis** | **83.3** (1) | **83.3** | 81.3 (2) | 82.0 |

Table 2: Ablations of LIGER, removing partitions (-Part), extensions (-Ext), and both. Best $s$ values inside parentheses.

## 5.3 ABLATIONS

We report ablations on each component of LIGER. Table 2 removes the partioning component and the extensions component. Partitioning improves performance on four tasks, and extensions improves performance on all tasks (13.1 points of lift on average from partitioning, 3.8 points from extensions). Combining both additionally offers the best performance on four tasks.

## 6 RELATED WORK

We present an abbreviated related work here. See Appendix A for an extended treatment.

Weak supervision frameworks typically model source accuracies to generate weak labels and then fine-tune an end model for generalization [Ratner et al., 2018, Bach et al.,

**Comparing Prompting Strategies** Next, we examine the impact of prompting strategies for **Spouse**. **Spouse** is a relation extraction dataset, where the task is to predict whether two entities in a sentence are married. Since there may be multiple entities in a sentence, **Spouse** contains multiple duplicate sentences in the dataset, with different labels. To alleviate this problem, we introduce a prompt "Are [person 1] and [person 2] spouses?" after the end of the sentence, where "[person 1/2]" are replaced by the names of the first/second entity in the sentence. We compare this prompting strategy against two others: appending the same prompt to the beginning of the sentence, and leaving the original sentence as-is, without any prompting.

Figure 2 (right) shows the performance and smoothness of each of these prompting methods. Adding the prompt to the end of the sentence results in the best performance and smoothest embeddings. Both methods perform better than leaving the sentence alone (the flat line is a result of multiple sentences with different labels having the same embedding).

2019, Khetan et al., 2018, Sheng et al., 2020, Fu et al., 2020, Zhan et al., 2019, Safranchik et al., 2020, Boecking and Dubrawski, 2019]. One framework models the end-to-end process all at once [Cachay et al., 2021], but requires training the end model at the same time—which is computationally expensive with large foundation models. Our work removes the fine-tuning step completely.

Our work is similar to transfer learning techniques, which adapt pretrained models for downstream tasks [Kolesnikov et al., 2020, Devlin et al., 2018]. Foundation models offer new requirements for transfer learning setting: when it is impossible to fine-tune the original models [Bommasani et al., 2021]. We build on approaches such as prompting [Lester et al., 2021, Brown et al., 2020], embedding search [Neelakantan et al., 2022], and adapters [Houlsby et al., 2019, Alain and Bengio, 2016].

# 7 CONCLUSION

We present LIGER, a system for fusing foundation models and weak supervision. We use embeddings to produce finer-grained estimates of weak source accuracies and improve weak source coverage. We prove a series of results on how the performance of this approach scales with the smoothness of the embeddings, and demonstrate LIGER on six benchmark NLP and video weak supervision datasets. We hope our work will encourage further work in combining foundation models and weak supervision and in utilizing the signal from foundation models to help with other tasks.

## Author Contributions

The first two authors contributed equally. Co-first authors can prioritize their names when adding this paper's reference to their resumes.

## Acknowledgements

We thank Fait Poms and Ravi Teja Mullapudi for helpful discussions. We thank Neel Guha, Megan Leszczynski, Vishnu Sarukkai, and Maya Varma for feedback on early drafts of this paper. We gratefully acknowledge the support of NIH under No. U54EB020405 (Mobilize), NSF under Nos. CCF1763315 (Beyond Sparsity), CCF1563078 (Volume to Velocity), 1937301 (RTML), and CCF2106707 (Program Synthesis for Weak Supervision); ARL under No. W911NF-21-2-0251 (Interactive Human-AI Teaming); ONR under No. N000141712266 (Unifying Weak Supervision); ONR N00014-20-1-2480: Understanding and Applying Non-Euclidean Geometry in Machine Learning; N000142012275 (NEPTUNE); NXP, Xilinx, LETI-CEA, Intel, IBM, Microsoft, NEC, Toshiba, TSMC, ARM, Hitachi, BASF, Accenture, Ericsson, Qualcomm, Analog Devices, Google Cloud, Salesforce, Total, the HAI-GCP Cloud Credits for Research program, the Stanford Data Science Initiative (SDSI), Department of Defense (DoD) through the National Defense Science and Engineering Graduate Fellowship (NDSEG) Program, Wisconsin Alumni Research Foundation (WARF), and members of the Stanford DAWN project: Facebook, Google, and VMWare. The U.S. Government is authorized to reproduce and distribute reprints for Governmental purposes notwithstanding any copyright notation thereon. Any opinions, findings, and conclusions or recommendations expressed in this material are those of the authors and do not necessarily reflect the views, policies, or endorsements, either expressed or implied, of NIH, ONR, or the U.S. Government.

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
