# OpenReview forum: "Shoring Up the Foundations: Fusing Model Embeddings and Weak Supervision"
_auai.org/UAI/2022/Conference — UAI 2022 Oral_

### Official Review · Reviewer_6oiR · 2022-04-10

**Q2(1) Originality/Novelty:** 4
**Q2(2) Significance/Impact:** 4
**Q2(3) Correctness/Technical Quality:** 4
**Q2(6) Clarity Of Writing:** 4
**Q6 Overall Score:** 7
**Q8 Confidence In Your Score:** 3

**Q1 Summary And Contributions:**

In this paper, the authors discuss how to utilize the foundation model to address the weak supervision problem. This is an interesting task to utilize the weak source classifier and the embedding from the foundation model. Hence, the authors first produce an estimation for each source, then they further improve the source converge by extending the source votes in embedding space. The authors evaluate the proposed method on six benchmark datasets and achieve a sound performance.

**Q2 Assessment Of The Paper:**

More detailed information regarding each of these aspects is given below:

**Q2(4) Quality Of Experiments (Optional):**

3: Good: The experimental evaluation is adequate, and the results convincingly support the main claims.

**Q2(5) Reproducibility:**

3: Good: Key resources (e.g., proofs, code, data) are available and key details (e.g., proofs, experimental setup) are sufficiently well-described for competent researchers to confidently reproduce the main results.

**Q3 Main Strengths:**

1.	The authors focus on an important problem, how to well utilize the foundation model. The proposed method is readily comprehensible and sound.
2.	The authors provide the generalization error bound for the proposed method.
3.	The proposed method outperforms six benchmarks with a large margin.


**Q4 Main Weakness:**

1.	The major concern of the proposed method is the label model. The authors claim that they model the P(y,\lambda|x) as a probabilistic graphical model, but it is unclear what the probabilistic graph looks like. It is suggested that the authors should provide the graphs.
2.	The proposed method is similar to ensemble learning since the proposed method also utilizes different sources/classifiers. It is suggested that the authors should introduce their connection in the related works.


**Q5 Detailed Comments To The Authors:**

Please refer to the aforementioned strengths and weaknesses.

**Q7 Justification For Your Score:**

I read the full paper, including the experiment details. I think this paper is interesting but  I am not an expert in this domain.

**Q9 Complying With Reviewing Instructions:**

1: Yes.

---

### Official Review · Reviewer_ws69 · 2022-04-11

**Q2(1) Originality/Novelty:** 3
**Q2(2) Significance/Impact:** 3
**Q2(3) Correctness/Technical Quality:** 3
**Q2(6) Clarity Of Writing:** 3
**Q6 Overall Score:** 6
**Q8 Confidence In Your Score:** 3

**Q1 Summary And Contributions:**

The paper aims to improve weak supervision with the help of embeddings obtained from the powerful foundation model.  It proposes LIGER to address two problems of weak supervision techniques.  Label model generalization error is presented and experiments on six benchmark NLP and video weak supervision tasks are conducted to demonstrate the advantages of the proposed method.

**Q2 Assessment Of The Paper:**

More detailed information regarding each of these aspects is given below:

**Q2(4) Quality Of Experiments (Optional):**

2: Fair: The experimental evaluation is weak: important baselines are missing, or the results do not adequately support the main claims.

**Q2(5) Reproducibility:**

2: Fair: Key resources (e.g., proofs, code, data) are unavailable but key details (e.g., proof sketches, experimental setup) are sufficiently well-described for an expert to confidently reproduce the main results.

**Q3 Main Strengths:**

1. An interesting improvement method for weak supervision techniques via combination with foundation model embedding space.
2. A theoretical analysis of generalization error bounds that characterize the impact of their approach.
3. Extensive experiments in six different tasks to demonstrate the advantages of the proposed method.

**Q4 Main Weakness:**

1. Inconsistent ideas and motivation in the paper, e.g. the title “shores up the foundations” while the paper addresses improving weak supervision techniques.
2. Some main issues are not clear enough, e.g. lack of the elaboration of the important partitioning component, the selection of the number of partitions, and the radius for extensions.
3. Some gaps exist in the experiments and the main claim, e.g. between prompting and smoothness.
4. Miss important experiments, e.g. the smoothness of text embedding space.

**Q5 Detailed Comments To The Authors:**

1. About consistency in the idea and motivation
    1. In the Introduction, “the fusion may enable each component to be improved”.  It seems that it is a little over-claimed here because the paper only addresses the problems of weak supervision techniques.  It should be better to rewrite it as the improvement of weak supervision techniques based on foundation model embedding.
    2. In the Introduction, the paper highlights the motivation and the problem “naively attempting to model per-point accuracies leads to noisy estimation”.  However, in Sec. 3 “FUSION ALGORITHM”, the statement is changed to “reduce the complexity, instead of estimating parameters per point”.  It seems that the paper does not try to tackle the problem it aims to.
    3. Sec. 2.2 ”LABEL MODEL”, it lacks some elaboration on the motivation of “we model p(y,\lambda|x) as a probabilistic graphical model”.  Besides, it seems that the paper directly estimates the posterior probability distribution without the joint distribution, which is inconsistent with the statement “in training, we must estimate the accuracy parameters of p(y,\lambda|x)”.
2. About clarity in the main method, e.g., :
    1. In Sec. 2.2 ”LABEL MODEL”, the statement “assume there are no dependencies” seems to conflict with the statement “Our choice of the base estimator can handle dependencies”.  The expression can be revised to be more consistent.
    2. In Sec. 2.2 ”LABEL MODEL”, the basis of conditional independence property: “An important property above is that \lambda_i \perp \lambda_j” is claimed.  It seems that this is an assumption rather than a property.  If it is a property, it is better to provide the possible design to attain it.
    3. In Sec. 2.2 ”LABEL MODEL”, the statement “parametric approaches are often not realistic” is unclear to the reviewer.
    4. In Sec. 3, “These estimates are done over the subsets” is unclear to the reviewer.
    5. In Sec. 4.1, the important partitioning component lacks sufficient elaboration. Moreover,  as stated in “As s increases, subset diameter d_C decreases”, it is not sure whether the claim is correct.
    6. In Algorithm 1,  how to estimate “p(\bar{\lambda}|C(x))” and p(\bar{\lambda}\ne 0|C_j) is unknown.  If the former one is estimated as “p(\lambda|C(x))” presented in “Local Parameter Estimation”, it should be explained why not to estimate it directly with p(\bar{\lambda}|y,C(x)) and p(y|C(x)).  Moreover, it is better to present Algorithm 2 in the main text accordingly rather than in the Appendix.
    7. In the second bullet term of the discussion on Theorem 1, the reviewer understands the parts of coverage and agreement among weak sources in the inequality of Variance.  However, it is unclear which part reveals “accuracy”.  It should be explained.
3. About the experiments
    1. While the paper has proved that the number of partitions and the radius for extensions control the performance, it does not provide their design methods. The choice of class balance is unknown too.
    2. The “Comparing Prompting Strategies” experiments lack persuasiveness since there is a gap between prompting and smoothness.
    3. Since the method heavily relies on the smoothness of embedding space, it is better to show experiments on text embeddings as well.
    4. WS-kNN and WS-Adapter lack descriptions, thus the reviewer is confused about the need for a train set and does not understand “Weak Sources and Dev Labels” at all.
    5. The measure of coverage should be explained, otherwise, it seems strange to the reviewer that it is a decimal in a single task.
    6. The statement “where LIGER labels are very accurate, we see an additional lift from the Adapters” seems to be improper since the highest lift in Table 1 is in the worst played task “Spouse”.
    7. In figure 2, two subfigures use different x-axis without explanation.
    8. The score type used in Table 2 is unclear to the reviewer.
4. Some other issues
    1. The partition function Z defined in section 2.2 “LABEL MODEL” seems to be improper as a denominator in equation (1).
    2. Sec. 2.1, the definition of “an unlabeled dataset” is unclear.

**Q7 Justification For Your Score:**

An interesting improved proposal for weak supervision techniques via foundation model embedding space with theoretical justification.

**Q9 Complying With Reviewing Instructions:**

1: Yes.

---

### Official Review · Reviewer_XaBQ · 2022-04-13

**Q2(1) Originality/Novelty:** 3
**Q2(2) Significance/Impact:** 3
**Q2(3) Correctness/Technical Quality:** 3
**Q2(6) Clarity Of Writing:** 2
**Q6 Overall Score:** 7
**Q8 Confidence In Your Score:** 3

**Q1 Summary And Contributions:**

The paper presents an algorithm, LIGER, that combines foundation
models with weak supervision by exploiting local smoothness of labels
and weak sources in embedding space.  The paper establishes
finite-sample error bounds of LIGER that scale in smoothness.  It
provides empirical evaluation of LIGER on several benchmark NLP and
video weak supervision tasks that shows promising performance of the
proposed technique.

**Q2 Assessment Of The Paper:**

More detailed information regarding each of these aspects is given below:

**Q2(5) Reproducibility:**

3: Good: Key resources (e.g., proofs, code, data) are available and key details (e.g., proofs, experimental setup) are sufficiently well-described for competent researchers to confidently reproduce the main results.

**Q3 Main Strengths:**

Detailed theoretical analysis with experimental validation.

**Q4 Main Weakness:**

Lack sufficient motivation for problems with weak supervision signals.

**Q5 Detailed Comments To The Authors:**

Class Balance in Eq. 1 needs to be defined. Also, need to clearly
define all the notations. The paper is somewhat presented in a
telegraphic style.

**Q7 Justification For Your Score:**

Overall it's a good paper. Detailed theoretical analysis, supported by experiments.

**Q9 Complying With Reviewing Instructions:**

1: Yes.

---

### Official Review · Reviewer_gWAe · 2022-04-13

**Q2(1) Originality/Novelty:** 3
**Q2(2) Significance/Impact:** 3
**Q2(3) Correctness/Technical Quality:** 3
**Q2(6) Clarity Of Writing:** 3
**Q6 Overall Score:** 8
**Q8 Confidence In Your Score:** 4

**Q1 Summary And Contributions:**

This paper introduces a new embedding-aware label model for weak
supervision: LIGER. LIGER uses embeddings from pre-trained foundation
models (e.g., GPT-3, CLIP) to learn more refined label model
parameters than existing approaches, and it uses label propagation in
this embedding space to extend the coverage of weak labels.

Empirical results show large improvements against strong weak
supervision baselines that also get access to the same embeddings.


**Q2 Assessment Of The Paper:**

More detailed information regarding each of these aspects is given below:

**Q2(4) Quality Of Experiments (Optional):**

3: Good: The experimental evaluation is adequate, and the results convincingly support the main claims.

**Q2(5) Reproducibility:**

3: Good: Key resources (e.g., proofs, code, data) are available and key details (e.g., proofs, experimental setup) are sufficiently well-described for competent researchers to confidently reproduce the main results.

**Q3 Main Strengths:**

- Clear writing
- Innovative and important idea
- Great empirical results showing improvement over strong baselines


**Q4 Main Weakness:**

Both minor points:

- Space is limited, but the description of the baselines is a bit
  rushed. E.g., It's not totally clear which baselines are label-model
  performance vs end-model performance. It seems like Table 1 WS-LM is
  the raw FlyingSquid performance, and the other WS- are end models
  trained on FlyingSquid labels using the foundation model embeddings?
  What happened to the WS- prefix in the right table?

- I think a lot of the theory could just be presented
  intuitively. It's not clear how much the details add given that you don't
  give a big empirical exploration of the tradeoffs you derive. "Suppose that
  data $x, y, \lambda$ follows the model in (1)." is a big assumption, so it would be great to have empirical verification of
  all the bias/variance tradeoffs you discuss, since I doubt the
  closed forms derived exactly match the tradeoffs in practice. Figure 2 is a great step towards this, though.

**Q5 Detailed Comments To The Authors:**


- typos on top of pg 16 (first equation)---in the second expectation
  you're missing a log and (I think) missing a $y$ in the subscripts
  to $\mathbb{E}$.

- formatting typo in last eqn on pg 16---\hat should just be on Pr

- Also, $f(x) = \log c/x$ is not the same $f(x)$ as the embedding.

- Seems like some major proof techniques are deferred to [1], but
  consulting Lemmas 4/6 of that work doesn't make it immediately obvious
  that we're in the same setting confronting the same problem. It
  would be nice to have a self-contained argument.

- Is it fair to say that the analysis for Section 4.1 largely replaces
  $x$ in [1] with $C(x)$ and the rest goes through? The theorem
  statements and techniques look quite similar.

- What happens in Theorem 1 if only one $C_i$ (say $C_1$) has points
  with $\lambda \ne 0$? That is, I have very small coverage, and it's
  all concentrated in $C_1$. Can't I still have $p_{min} > 0$ in this
  case? But it doesn't make sense that the bound could tell me
  anything nontrivial in this case.  Is there an assumption that rules
  this case out? The irreducible error is conditioned on $\lambda$
  *and* $x$, so I'm imaging a case where $y = g(x)$ and this term is 0.

- When the development set is really large (Spouse, Commercial), why
  not use ~half of it to learn the label model parameters directly
  from labeled data instead of bothering with complicated
  FlyingSquid/DP/MeTaL style things?  Especially with a
  DP/Dawid-Skene-style setup where there's basically only ~num-classes
  relevant parameters, it shouldn't take much labeled data.

- Different line-of-work, but it might be good to mention that others
  have looked at using foundation models as a *source* of weak
  supervision / as an initial hypothesis in co-training.  E.g., [2] and
  followup work.

- _I think a lot of the theory could just be presented intuitively._
E.g., Figure 2: Basketball has pretty small
  changes in Lipschitzness but huge changes in performance between
  embeddings. Spouse has huge changes in Lipschitzness but much
  smaller changes in performance.  This isn't really following the
  closed-form bias formula in the bounds...so perhaps the theory
  (while very interesting) could be lighter on detail in the main
  text? Replacing some of the theory details with more explorations like Figure 2 would make the paper easier to read.

- Will code extending FlyingSquid with this technique be released? I
  didn't see any mention of code release in the text.


[1] Mayee Chen, Benjamin Cohen-Wang, Stephen Mussmann, Frederic Sala,
and Christopher Re. Comparing the value of labeled and unlabeled data
in method-of-moments la- tent variable estimation. In Proceedings of
The 24th International Conference on Artificial Intelligence and
Statistics, 2021

[2] Wang, Shuohang, et al. "Want To Reduce Labeling Cost? GPT-3 Can
Help." Findings of the Association for Computational Linguistics:
EMNLP 2021. 2021.

**Q7 Justification For Your Score:**

Great idea, strong experiments, clear presentation.

**Q9 Complying With Reviewing Instructions:**

1: Yes.

---

### Decision · Program_Chairs · 2022-05-15

**Decision:**

Accept (Oral)

**Comment:**

Meta Review: This paper proposes LIGER to better use the foundation model in a weakly-supervised setting. LIEGER has two key components: 1) a finer estimator of weak source quality by dividing the embedding space and learning the source accuracy for each part, 2) source voting expansion  in the embedding space.

The reviewers agree that the proposed work is innovative and makes a significant technical contribution in using pretrained models under weak supervision. There were some concerns raised by reviewers, but most of them have been well addressed by the authors.  I recommend acceptance of this paper given its novelty and significant technical contribution.